# Ginseng Saponin Enriched in Rh1 and Rg2 Ameliorates Nonalcoholic Fatty Liver Disease by Inhibiting Inflammasome Activation

**DOI:** 10.3390/nu13030856

**Published:** 2021-03-05

**Authors:** Feng Wang, Jeong-Su Park, Yuanqiang Ma, Hwan Ma, Yeo-Jin Lee, Gyu-Rim Lee, Hwan-Soo Yoo, Jin-Tae Hong, Yoon-Seok Roh

**Affiliations:** College of Pharmacy and Medical Research Center, Chungbuk National University, Cheongju, Chungbuk 28160, Korea; wang979253414@gmail.com (F.W.); 6318js@gmail.com (J.-S.P.); Yuangqiangma123@gmail.com (Y.M.); akghks5065@gmail.com (H.M.); leeyeojin53@gmail.com (Y.-J.L.); ssonhm117@gmail.com (G.-R.L.); jinthong@chungbuk.ac.kr (J.-T.H.); yoohs@chungbuk.ac.kr (H.-S.Y.)

**Keywords:** NAFLD, ginsenosides, inflammasome, mitophagy

## Abstract

Nonalcoholic fatty liver disease (NAFLD) is becoming one of the most common chronic liver diseases in the world. One of the features of NAFLD is hepatic fat accumulation, which further causes hepatic steatosis, fibrosis, and inflammation. Saponins, the major pharmacologically active ingredients isolated from *Panax notoginseng*, contain several ginsenosides, which have various pharmacological and therapeutic functions. However, the ginsenoside-specific molecular mechanism of saponins in NAFLD remains unknown. This study aimed to elucidate the effects of ginseng saponin extract and its ginsenosides on hepatic steatosis, fibrosis, and inflammation and their underlying action mechanism in NAFLD. Mice were fed a fast food diet (FFD) for 16 weeks to induce NAFLD and then treated with saponin extract (50 or 150 mg/kg) for the remaining nine weeks to determine the effects of saponin on NAFLD. Saponin extract administration significantly alleviated FFD-induced hepatic steatosis, fibrosis, and inflammation. Particularly, saponin extract, compared with conventional red ginseng, contained significantly increased amounts of ginsenosides (Rh1 (10.34-fold) and Rg2 (7.1-fold)). In vitro Rh1 and Rg2 treatments exerted an anti-steatotic effect in primary hepatocytes, an antifibrotic effect in hepatic stellate cells, and anti-inflammatory and pro-mitophagy effects in immortalized mouse Kupffer cells. Mechanistically, saponin extract alleviated lipopolysaccharide-induced NLRP3 inflammasome activation by promoting mitophagy. In conclusion, saponin extract inhibited inflammation-mediated pathological inflammasome activation in macrophages, thereby preventing NAFLD development. Thus, saponin extract administration may be an alternative method for NAFLD prevention.

## 1. Introduction

Nonalcoholic fatty liver disease (NAFLD) appears to be a major public health concern in Western countries and the Asia-Pacific region [1,2]. NAFLD includes simple steatosis and nonalcoholic steatohepatitis (NASH) and can further develop into cirrhosis or hepatocellular carcinoma [3]. Although most NAFLD patients remain asymptomatic, 20% of them progress to NASH [4]. However, the mechanism of progression from NAFLD to NASH remains unclear [5]. “Two hits” were regarded as the major drivers for NASH development from NAFLD. The proposed “first hit” was hepatocyte lipid accumulation and the associated lipotoxicity-induced mitochondrial abnormalities. The proposed “second hit” was a multifactorial process, which involved proinflammatory cytokines, reactive oxygen species (ROS), and lipid peroxidation [6]. Additionally, there is persistent hepatocyte inflammation, leading to hepatic steatosis, chronic inflammation, and hepatocyte damage, in NASH [7]. Elucidating the inflammatory molecular mechanism underlying NAFLD has been described as the most important endeavor for development of therapeutic regimens.

The inflammasome is a cytoplasmic multiprotein complex, comprised of the sensor proteins, such as the nucleotide-binding domain and leucine-rich repeat-containing (NLR), and PYHIN proteins, including NLRP1, NLRP3, and NLRC4, and absent in melanoma 2 (AIM2) [4,8]. The NLRP3 inflammasome can recognize various substances, including damage-associated and pathogen-associated molecular patterns, and activate the innate immune response, such as in Kupffer cells (KCs), thereby triggering caspase-1 activation and subsequent interleukin (IL)-18 and IL-1β maturation [9,10,11,12,13]. IL-18 and IL-1β have been recognized as proinflammatory cytokines associated with tumorigenesis [14]. Substantial evidence on the association of inflammasome activation and IL-1β secretion with NAFLD and NASH development has been presented [15,16,17]. Based on this evidence, NLRP3 and the inflammasome pathways must be tightly controlled, and the regulation of inflammasome activity must be further explored.

The mitochondrion is the major energy-producing organelle that plays an essential role in lipid metabolism regulation, hepatic cellular redox reactions, and cell death. Mitochondrial dysfunction is associated with both chronic and acute liver disease [18]. Accumulating evidence has indicated that selective autophagy—specifically, mitophagy—when impaired, contributes to alcoholic liver disease, NAFLD, drug-induced liver injury, hepatic ischemia-reperfusion injury, viral hepatitis, and liver cancer and plays a vital role in the regulation of liver homeostasis [19,20,21,22,23]. Long-term inflammation increases mitochondrial ROS (mtROS) production, causing oxidative stress and, subsequently, mitochondrial dysfunction. Damaged mitochondria accumulation releases redundant mtROS or mitochondrial DNA (mtDNA) into the cytosol, triggering NLRP3 inflammasome activation [24,25]. The mitochondrial dynamics must be balanced well with mitophagy and mitochondrial biogenesis to maintain mitochondrial homeostasis [26]. Under mitochondrial dysfunction states, mitophagy is activated to clear the damaged mitochondria to restore normal mtROS levels and further inhibit NLRP3 inflammasome activation. Thus, mitophagy maintains mitochondrial homeostasis and regulates innate immune responses [27].

Various saponins have been identified as the major pharmacologically active ingredients isolated from *Panax notoginseng* [28,29]. Red ginseng (RG; steamed and dried root of *Panax* sp.) contain many compounds, including saponins, polysaccharides, polyalkenes, sterols, and essential oils. The major bioactive constituents found in RG are the triterpenae saponins, also referred to as ginsenosides or panaxosides [21]. The most important compounds of saponins are ginsenosides, which have various pharmacological and therapeutic functions. According to the different carbohydrate moieties at the C3, C6, and C20 positions, ginsenosides can be divided into protopanaxatriols (Rg1, Re, Rg2, Rh1, and Rf); protopanaxadiols (Rb1, Rb2, Rd, Rg3, and Rh2); and oleanolic acid derivatives (Ro) [30,31]. Substantial evidence has suggested that ginsenosides have antioxidant, anti-inflammatory, anti-apoptosis, and neuroprotective properties [30]. Ginsenoside Rg1 may provide protection against NAFLD by regulating lipid peroxidation, inflammasome activation, and endoplasmic reticulum (ER) stress [32]. Ginsenoside Rb2 alleviates lipid accumulation by restoring autophagy to improve the tolerance of NAFLD and glucose [33]. Ginsenoside M1 exerts protective effects against ER stress-induced apoptotic damage, insulin resistance, and lipogenesis; thus, it may be a potential therapeutic agent for NAFLD [34]. Even though it has been suggested that ginsenosides exert protective effects against NAFLD, the regulation and action mechanisms of ginsenosides Rg2 and Rh1 have not been clearly illustrated to date. This current study aimed to investigate the effects of saponin extract and its ginsenosides on hepatic steatosis, fibrosis, and inflammation and elucidate their underlying mechanisms of action in an NAFLD mouse model.

## 2. Materials and Methods

### 2.1. Animal Experiment and Ethical Approval

Male C57BL/6 mice (8 weeks old) were purchased from Samtako Bio Korea (Osan, Korea). All experimental procedures involving animals were approved by the Chungbuk National University Institutional Animal Care and Use Committee (IACUC). The protocol was approved by the IACUC of the Laboratory Animal Research Center at Chungbuk National University, Cheongju, Korea (Ethical approval No. CBNUA-1203-18-02). Mice were housed in a facility under a 12-h light/dark cycle at 21 ± 2 °C and then transferred to a specific-pathogen-free facility. High pressure was maintained in the experimental room to prevent contamination of the facility. Mice were then randomly divided into the following four experimental groups: the (1) normal chow diet (NCD)-fed; (2) fast food diet (FFD; 0.2% cholesterol, RD (Research Diets) Western diet, open source diets, 40% calories, and fructose (23.1 g/L) and glucose (18.9 g/L) solutions)-fed (Table 1); (3) FFD-fed and low-dose saponin extract (50 mg/kg)-administered; and (4) FFD-fed and high-dose saponin extract (150 mg/kg)-administered groups. All mice were fed either NCD or FFD for 16 weeks and then treated with either water or saponin extract (50 or 150 mg/kg) once daily by oral gavage for the remaining 9 weeks.

### 2.2. Chemicals

Dulbecco’s Modified Eagle’s Medium (DMEM), Corning^®^, Glendale, CA, USA; Fetal Bovine Serum (FBS), Corning^®^, Glendale, USA; M199 medium, Corning^®^, CA, USA; Penicillin-Streptomycin solution 100× (PS), Corning^®^, CA, USA; TB Green Premix Ex Taq II, TAKARA, Shiga, Japan; Lipopolysaccharide (LPS), Sigma-Aldrich, St. Louis, MO, USA; Carbonyl Cyanide m-Chlorophenylhydrazone (CCCP), Sigma-Aldrich, St. Louis, USA; Palmitic Acid, Sigma-Aldrich, St. Louis, USA; Oil red O, Sigma-Aldrich, St. Louis, USA; Pierce™ BCA Protein Assay Kit, Thermo Scientific, Waltham, MA, USA; Nuclear factor of kappa light polypeptide gene enhancer in B-cells inhibitor, alpha (IkBα), Santa Cruz, CA, USA; Phosphorylated-IkBα, Santa Cruz, CA, USA; Heat shock protein 60, Santa Cruz, CA, USA; Translocase of outer mitochondrial membrane 40 homolog, Santa Cruz, CA, USA; Mitochondrial import inner membrane translocase, Santa Cruz, CA, USA; Actin antibodies, Santa Cruz, CA, USA; Primer Script^TM^ RT Reagent Kit with gDNA Eraser, Shiga, Japan; ATP, Sigma-Aldrich, St. Louis, USA; Metformin, Sigma-Aldrich, St. Louis, USA; NLRP3 inhibitor, Selleckchem, Houston, TX, USA; Ginsenoside Rh1, Merck company, Kenilworth, NJ, USA and Ginsenoside Rg2, Merck company, NJ, USA;

### 2.3. Preparation for RG and Saponin Extract

Fresh ginseng roots were prepared and processed by steaming and drying to make red ginseng in the red ginseng manufacturing factory of Korea Ginseng Corporation (Buyeo, Chung-nam, Korea). Washed fresh ginseng roots were steamed for 4 h while slowly raising its temperature from 50 °C to 98 °C and then, firstly, dried at 60~70 °C for 15 h. Thereafter, a secondary drying process was performed in a closed chamber at 50 °C for 5 days to result in the red ginseng roots (RG). To prepare red ginseng extract, the root was sequentially extracted 7 times at 87 °C for 12 h with distilled water. The extracted water was combined followed by a filtering and concentrating process.

Non-saponin fraction and saponin fraction of the red ginseng were prepared by adsorption chromatography by the methods below with Dion HP20 (Mitsubishi Chemical Industries, New York, NY, USA) using red ginseng extract. The red ginseng extract was diluted to 10% in water and then filtered. The diluted solution was subjected to HP20 resin for adsorption then eluted using water, 30% ethanol in water, and 95% ethanol in water, sequentially. The first two fractions (H_2_O and 30% EtOH in water) were combined, concentrated, and spray-dried to produce the non-saponin fraction (NS-RG). The last fraction (95% EtOH in water) was concentrated and spray-dried to produce the saponin fraction (S-RG).

### 2.4. Cell Culture

Immortalized mouse KCs (ImKCs) were purchased from Applied Biological Materials Inc. (Richmond, BC, Canada), which were the second passage, and cultured in Dulbecco’s Modified Eagle’s Medium (DMEM) supplemented with 10% fetal bovine serum (FBS) in a 5% CO_2_ atmosphere at 37 °C. To demonstrate the anti-inflammatory effects of saponin extract and ginsenosides Rh1 and Rg2 and measure the relevant gene mRNA levels, ImKCs (2.5 × 10^5^ cells/mL) were incubated with lipopolysaccharide (LPS, 125 ng/mL), saponin extract (31.25 or 62.5 µg/mL), or red ginseng (62.5 µg/mL) and Rh1 or Rg2 (62.5 µg/mL) for 6 h at 37 °C. To measure the relevant gene protein levels, ImKCs (5 × 10^5^ cells/mL) were treated with saponin extract at the same concentration 3 h before LPS treatment and then cultured for 30 min at 37 °C. To detect the effects of Rh1 and Rg2 on mitophagy, ImKCs (5 × 10^5^ cells/mL) were treated with carbonyl cyanide m-chlorophenylhydrazone (CCCP, 10 µM) and Rh1 or Rg2 for 24 h. To detect the effects of Rh1 and Rg2 on mtROS production, ImKCs (2.5 × 10^5^ cells/mL) were treated with LPS (400 ng/mL) and Rh1 or Rg2 for 3 h.

Primary hepatocytes were isolated from the C57BL/6 mice, as previously described [35]. Hepatocytes were resuspended in complete M199 medium. To detect lipid accumulation, primary hepatocytes (1 × 10^5^ cells/mL) were incubated with Palmitic Acid (PA, 200 µM) and saponin extract (31.25 or 62.5 µg/mL) for 24 h at 37 °C, and BODIPY staining was performed. To detect the relevant gene mRNA levels, primary hepatocytes (5 × 10^5^ cells/mL) were treated with saponin extract at the same concentration 3 h before PA treatment and then cultured for 9 h at 37 °C. Hepatic stellate cells (HSCs) were isolated from the C57BL/6 mice and resuspended in DMEM containing 10% FBS. To confirm the effects of saponin extract on fibrosis, HSCs (2 × 10^4^ cells/mL) were incubated with saponin extract for 1 and 3 d at 37 °C.

### 2.5. Biochemical Analysis

Serum was collected from blood samples (centrifugation at 13,000 rpm for 15 min), and a biochemistry analyzer (Green Cross LabCell, Yongin, Korea) was used to assay the alanine aminotransferase (ALT) and cholesterol levels.

### 2.6. Histological Analysis of the Liver

Excised liver tissues were thoroughly cleaned, fixed in formalin, and embedded in paraffin. To visualize the lipid deposition, tissue blocks were cut into 5-μm-thick sections and stained with hematoxylin and eosin. To detect collagen deposition, 5-μm-thick sections were stained with Sirius Red. To assess hepatic steatosis, liver tissues were embedded in an optimum cutting temperature compound (Tissue-Tek; Sakura Finetek, Tokyo, Japan) were stained with Oil red O (American MasterTech, Lodi, CA, USA), according to the manufacturer’s instructions. For a quantitative analysis of lipid deposition, collagen deposition, and hepatic steatosis, the scanned liver sections were captured using a DMi8 (Leica Camera, Wetzlar, Germany) at 200-fold magnification, and the several positive areas were measured using LAS X (Leica Camera) and ImageJ software (National Institutes of Health, Bethesda, MD, USA).

### 2.7. Western Blot Analysis

Differently treated cells were washed twice with Phosphate-buffered saline (PBS), collected, and lysed on ice using Radioimmunoprecipitation assay buffer (RIPA) for 30 min. Liver tissues were directly homogenized on ice using RIPA buffer for 30 min. Cell and tissue lysates were subjected to centrifugation at 13,000 rpm at 4 °C for 15 min. Protein concentration of the supernatants was detected using the BCA Protein Assay Kit (Thermo Fisher Scientific Inc., Waltham, CA, USA). Then, an equal amount of protein was separated by 10% or 15% SDS-PAGE and transferred to polyvinylidene difluoride membranes. The membranes were blocked with 5% skim milk at room temperature for 1 h. Proteins were incubated with specific IkBα (A1187), Phosphorylated-IkBα (AP0731), Heat shock protein 60 (HSP60 and A0969), Translocase of outer mitochondrial membrane 40 homolog (Tom40 and SC-365467), Mitochondrial import inner membrane translocase (Tim23 and A8688), and Actin (#3700) antibodies, diluted to 1:1000, at 4 °C overnight and then horseradish peroxidase-conjugated secondary antibodies at room temperature for 1 h. All antibodies were diluted in tris-buffered saline/Tween containing 2% bovine serum albumin.

### 2.8. Quantitative Real-Time Polymerase Chain Reaction (PCR) Analysis

Cells and liver tissues were homogenized in RiBoEx (Cat. No. 301-001), and total RNA was reverse-transcribed to complementary DNA (cDNA) using the PrimerScript^TM^ RT Reagent Kit with Genomic DNA (gDNA) Eraser (Cat. No. RR037A). PCR was performed using the CFX Connect Real-Time PCR Detection System (Bio-Rad, Hercules, CA, USA). The targeted gene expression was normalized to glyceraldehyde 3-phosphate dehydrogenase (GAPDH, internal control) expression. The specific primer sequences are listed in Table 2.

### 2.9. Cytotoxicity

ImKCs were plated in filtered DMEM supplemented with 10% FBS in 12-well plates (2.5 × 10^5^ cells/mL) for 12 h. The medium was then replaced with fresh medium (without FBS) supplemented with saponin extract (Rh1 and Rg2) at a different concentration with or without LPS. The medium was incubated for 24 h at 37 °C and then collected for the required assays. PA cytotoxicity towards primary hepatocytes was assessed, as previously described [35].

### 2.10. Statistical Analysis

All data are expressed as the means ± standard error of the mean (sem) from at least three independent experiments. GraphPad Prism 8 software (GraphPad Software Inc., San Diego, CA, USA) was used to perform statistical analysis by one-way analysis of variance. *p*-values < 0.05 were considered statistically significant.

## 3. Results

### 3.1. Saponin Extract Improved FFD-Induced Hepatic Steatosis

To investigate whether saponin extract treatment regulated the FFD-induced hepatic lipid accumulation in mice, mice were fed an FFD for 16 weeks to induce obesity, which is a relevant feature of NAFLD. Mice were weighed daily, and the body weights of FFD-fed mice, compared to that of NCD-fed mice, were significantly increased, as expected (Appendix A). Surprisingly, the weight (Figure 1A) and size (Figure 1B) of the liver was significantly attenuated in the low- or high-dose saponin extract (SapL and SapH, respectively)-administered groups, compared with the vehicle group, but no effects on body weight, liver-to-body weight ratio, and food uptake were observed from saponin extract administration (Appendix A). Additionally, the cholesterol levels were significantly reduced in the SapH-treated group, compared with the vehicle group (Figure 1C). Furthermore, saponin extract administration significantly alleviated the FFD-induced increase in lipid droplet accumulation (Figure 1D,E). To explore the beneficial effects of saponin extract treatment on hepatic lipid metabolism further, the expression of various genes involved in lipogenesis and fatty acid oxidation were examined. Saponin extract treatment reversed the FFD-induced elevation in the expression levels of lipogenesis-related genes, including fatty acid synthase (FASN) and MLX-interacting protein-like (MLXIPL), to the levels in NCD-fed mice (Figure 1F,G). The expression of fatty acid oxidation-related genes, including carnitine palmitoyltransferase I alpha (CPT-1α) and peroxisome proliferator-activated receptor alpha (PPARα), were upregulated in FFD mice and further enhanced on saponin extract administration (Figure 1H,I). Taken together, these results indicated that saponin extract administration ameliorated hepatic steatosis by regulating the expression of genes involved in hepatic lipid metabolism in vivo.

To clarify the effects of saponin extract on hepatic steatosis in vitro, PA was used to induce lipid accumulation in mouse primary hepatocytes. Lipid accumulation was assessed by BODIPY staining (Figure 2A) and evaluated by quantification of the BODIPY/DAPI (diamidino-2-phenylindole)-positive areas (Figure 2B). Both SapL and SapH treatments significantly decreased the PA treatment-induced lipid accumulation. Based on these results, the various hepatic lipid metabolism-involved genes, including lipogenesis-related genes (MLXIPL (Figure 2C) and sterol regulatory element-binding protein (SREBP)-1c (Figure 2D)) and fatty acid oxidation-related genes (CPT-1α (Figure 2E) and PPARα (Figure 2F)), were detected. As shown in Figure 2F, saponin extract treatment restored the PA treatment-induced increase in SREBP-1c and MLXIPL expression and further enhanced the PA treatment-induced increase in CPT-1α and PPARα expression. According to these results, saponin extract exerted beneficial effects on hepatic steatosis by alleviating it in vivo and in vitro.

### 3.2. Saponin Extract Alleviated FFD-Induced Steatofibrosis

Next, we investigated the effects of saponin extract treatment on fibrosis in the pathogenesis of FFD-induced NAFLD. According to the Sirius Red staining results, saponin extract treatment reduced the FFD-induced collagen deposition (Figure 3A,B). Consistently, we found that saponin extract treatment significantly decreased the mRNA levels of various fibrosis-related genes, including collagen type IV (COL4) (Figure 3C) and lysyl oxidase (LOX) (Figure 3D). To examine these antifibrotic effects further, we utilized the in vitro fibrotic model using primary HSCs. Saponin extract significantly decreased the culture-induced expression of fibrosis-related genes, including tissue inhibitor of metalloproteinase (TIMP) (Figure 3E), COL1 (Figure 3F), LOX (Figure 3G), and COL3 (Figure 3H). Taken together, these results indicated that saponin extract administration ameliorated hepatic fibrosis.

### 3.3. Saponin Extract Was Found to Contain High Contents of Ginsenosides Rh1 and Rg2, Which Exerted Anti-Inflammatory Effects

To clarify whether saponin extract exerted anti-inflammatory effects, we measured the expression levels of various inflammatory genes in the liver. As shown in Figure 4A–D, saponin extract significantly reduced FFD-induced hepatic arginase 1 (ARG1), C-C motif ligand 2 (CCL2), CCL4, and C-X-C motif ligand 2 (CXCL2) expression and the tissue tumor necrosis factor (TNF)-α level (Figure 4E), indicating that saponin extract exerted anti-inflammatory effects on FFD-induced steatohepatitis. Additionally, saponin extract treatment provided protection against FFD-induced liver injury by reducing the serum ALT level (Figure 4F). To examine the anti-inflammatory activity of saponin extract in vitro, we utilized the LPS-induced inflammation model using ImKCs. Additionally, we compared the anti-inflammatory activity of saponin extract and red ginseng. As shown in Figure 4G–I, the LPS treatment significantly altered ARG1, CCL2, and IL-1β mRNA expression, but both saponin extract and red ginseng treatments, compared with the LPS treatment, significantly decreased the inflammatory response. Moreover, saponin extract exerted greater anti-inflammatory effects than conventional red ginseng. This result prompted us to investigate whether saponin extract contained more functional compounds than red ginseng. As shown in Table 3 and Figure 4J, the saponin extract contained significantly increased amounts of several ginsenosides. Many of the upregulated ginsenosides, such as Rg1, Rg3, Rb1 [32,36], Rb2 [33], and Rd [35], were identified to be associated with NAFLD/NASH. In contrast, the role of Rh1 and Rg2 in hepatic pathophysiology remains unknown. Previous reports have hinted at a possible role of Rh1 and Rg2 in the inflammation of other organs, such as the kidneys and colon [37,38]. Thus, we then examined the anti-inflammatory effects of ginsenosides Rh1 and Rg2 and found that both ginsenosides significantly alleviated the LPS-induced expression of inflammatory genes, including TNF-α (Figure 4K), CCL2 (Figure 4L), ARG1 (Figure 4M), and IL-10 (Figure 4N), in ImKCs.

### 3.4. Rh1 and Rg2 Treatment Inhibited NLRP3 Inflammasome Activation

It is well-known that activation of the NF-kB pathway activates the IkB kinase complex, which is then degraded to release NF-kB and initiate the inflammatory response [39]. As shown in Figure 5A, saponin extract exerted stronger anti-inflammatory effects, as demonstrated by IkB degradation and phosphorylation. The NF-kB pathway is involved in the regulation of the inflammasome, which contributes to the initiation and development of inflammatory diseases [40]. Additionally, we found that the IL-1β production, mRNA expression in vivo and in vitro, and tissue level were reduced (Figure 5B–D). Thus, we focused on the inflammasome, which has been suggested as the major molecule involved in IL-1β secretion [41]. Saponin extract treatment alleviated the FFD-induced activation of the NLRP3 inflammasome and Aim2 inflammasome (Figure 5E,F). To investigate the effects of ginsenosides Rh1 and Rg2 on the inflammasome, we tested whether Rh1 and Rg2 affected both the expression and activation of the NLRP3 inflammasome. As shown in Figure 5G–H, Rh1 and Rg2 significantly suppressed the LPS+ATP-induced activation of the NLRP3 and AIM2 inflammasomes and, subsequently, the NLRP3-mediated secretion of mature IL-1β (Figure 5I). Finally, we found that expression of the inflammatory genes was not inhibited by Rh1 and Rg2 treatment in the inflammasome inhibitor-treated cells, compared with the vehicle group (Figure 5J–O), indicating that the anti-inflammatory activity of these ginsenosides depended on inflammasome signaling.

### 3.5. Ginsenosides Rh1 and Rg2 Alleviated NLRP3 Activation by Promoting Mitophagy

To elucidate the molecular mechanism of Rh1 and Rg2-mediated NLRP3 inhibition, we examine the effect of the ginsenosides on the mitochondria quality control, which was shown to involve the regulation of the NLRP3 inflammasome [42]. As shown in Figure 6A, the Rh1 and Rg2 treatment significantly decreased LPS-induced mtROS production and further alleviated mitochondrial damage. Since the alteration in mitophagy contributes to the accumulation of severely damaged and dysfunctional mitochondria, possibly promoting liver inflammation and NASH development [43], we analyzed the effects of Rh1 and Rg2 on mitophagy using the Mt-Keima system [44]. Rh1 and Rg2 enhanced CCCP-induced mitophagy compared with the control group (Figure 6B). As shown in Figure 6C, ginsenosides Rh1 and Rg2 significantly enhanced the CCCP-induced degradation of mitochondrial proteins such as Tom40, Tim23, and HSP60. Furthermore, ginsenoside Rh1 and Rg2 treatments also heightened the mitophagy induced by metformin (Figure 6D). We next sought to determine whether Rh1 and Rg2-increased mitophagy regulates inflammasome activation. As shown in Figure 6E–H, the metformin treatment improved IL-1β and NLRP3 inflammasome activation induced by a cotreatment with LPS and ATP, which was further decreased by Rh1 and Rg2. These results indicated that ginsenosides Rh1 and Rg2 improved the mitophagy to maintain mitochondrial homeostasis under stress and, subsequently, inhibit NLRP3 inflammasome activation.

## 4. Discussion

With the development of society, the prevalence of obesity, which induces diabetes, cerebrovascular diseases, high blood pressure, NAFLD, and cancer, in developed and developing countries is increasing every year [45]. Therefore, obesity is considered as the key risk factor for chronic liver diseases, especially NAFLD, which is a spectrum of liver pathologies encompassing the progression of steatosis, in the initial early stage, to steatohepatitis and fibrosis [46]. In recent years, NAFLD alleviation and treatment has mainly been aimed at decreasing lipid accumulation and synthesis and the inflammatory response and increasing fatty acid decomposition in fatty liver cells to restore normal metabolism [47,48]. Although NAFLD has been extensively studied, there is no approved drug for NAFLD treatment.

In recent years, many researchers have reported that some pure compounds or plant extracts may improve NAFLD [49,50,51]. Rutin could suppress hepatic lipid levels and oxidative injury to improve NAFLD in mice [52], ginsenoside Rg1 could effectively ameliorate hepatic steatosis and inflammation possibly via the AMPK/NF-kB pathway [53], and ginsenoside Rb1 alleviated high-fat diet-induced hepatocyte apoptosis in mice [54]. Red ginseng has been used as a natural medicine for various diseases in Asian countries for many years. In our study, saponin extract, extracted from *Panax notoginseng*, exerted better protective effects against inflammation than red ginseng. Additionally, saponin extract contained significantly increased amounts of several ginsenosides, such as Rh1 and Rg2. Although many studies have identified upregulated NAFLD/NASH-associated ginsenosides, such as Rg1, Rb1, Rg3 [32,36], Rd [35], and Rb2 [33], the effects of ginsenosides Rh1 and Rg2 on FFD-induced NAFLD and their potential mechanisms have not yet been explored. In the current study, the anti-inflammatory effects of saponin extract were investigated in vivo and in vitro. We found that ginsenosides Rh1 and Rg2 inhibited NLRP3 inflammasome activation and alleviated the inflammatory response by promoting mitophagy.

Substantial evidence suggests that saponin extract treatment mitigates NAFLD. FFD-fed mice are suitable for not only observing the effects of obesity but, also, studying NAFLD [55]. In our study, mice were fed with FFD for 16 weeks to induce NAFLD and then treated with saponin extract for the remaining nine weeks. Then, a biochemical analysis and histopathological examination were conducted in vivo. Saponin extract administration attenuated FFD-induced hepatic steatosis, inflammation, and fibrosis. Additionally, the serum levels of ALT, which is the standard indicator of liver function and is used to reflect hepatic injury and inflammation in patients with chronic liver diseases [56] and cholesterol homeostasis dysfunction, were documented in NAFLD [57]. We found that FFD-induced obese mice showed higher ALT and cholesterol levels, and the saponin extract treatment significantly decreased this increase in the serum ALT and cholesterol levels, thereby alleviating hepatic inflammation and damage. To elucidate the effects of saponin extract treatment on NAFLD further, we used PA to induce lipid accumulation and LPS to activate the inflammatory response and culture-treated the cells to stimulate fibrosis. According to our results, the saponin extract treatment significantly alleviated PA-induced lipid accumulation by decreasing the lipogenesis-related gene expression and increasing the fatty acid oxidation-related gene expression, as demonstrated by BODIPY staining of the primary hepatocytes. Additionally, the saponin extract treatment significantly improved the culture-induced fibrosis by reducing the fibrosis-related gene expression. Moreover, saponin extract decreased the lipid accumulation and fibrosis in a concentration-dependent manner. We also compared the anti-inflammatory effects of red ginseng and saponin extract. Both the red ginseng and saponin extract treatments significantly blocked the LPS-induced inflammatory cytokines. Interestingly, saponin exerted stronger anti-inflammatory effects than red ginseng, possibly owing to its high-content ginsenosides, Rh1 and Rg2. Both ginsenosides significantly decreased the LPS-induced inflammatory cytokines. Interestingly, we also observed the reduction in IL-1β production, mRNA expression in vivo and in vitro, and tissue levels. Accumulating evidence has determined that IL-1β is deeply involved in NAFLD and NASH pathogenesis [58,59]. Additionally, NLRP3 inflammasome activation, which markedly increases pro-IL-1β secretion in the liver, was associated with NASH [60]. Thus, we determined whether ginsenosides Rh1 and Rg2 exerted anti-inflammatory effects by inhibiting NLRP3 inflammasome activation.

Notably, our study demonstrated that NLRP3 inflammasome activation and NLRP3-mediated IL-1β maturation in the liver were dramatically suppressed in saponin extract-administered mice, compared with NCD-fed mice. Interestingly, the ginsenosides Rh1 and Rg2 were observed to exert similar effects on NLRP3 inflammasome activation and pro-IL-1β secretion in LPS+ATP-treated ImKCs. However, in contrast to the reduction in NLRP3 inflammasome activation and IL-1β secretion when Rh1 and Rg2 cells were treated with only LPS+ATP, no significant changes in NLRP3 inflammasome activation and IL-1β secretion were observed when Rh1 and Rg2 cells were treated with an NLRP3 inflammasome inhibitor. These results suggest that the anti-inflammatory effects exerted by ginsenosides Rh1 and Rg2 depend on NLRP3 inflammasome signaling.

Damaged mitochondria accumulation may activate the NLRP3 inflammasome by releasing mtROS or mtDNA into the cytosol [24,25]. Mitochondrial-selective degradation controls the mitochondrial number and health to regulate homeostasis via autophagy. Mitophagy, the selective elimination of excess or damaged mitochondria, is a process that links mitochondria and autophagy [61,62]. Accumulating evidence suggests that mitophagy inhibits inflammasome activation [24,25,42]. We employed the Mt-Keima system to measure the mitophagy and found that mitophagy was enhanced by Rh1 and Rg2. Additionally, the degradation of the mitochondrial proteins was measured to assess the mitophagy with respect to the biochemical aspects [63,64]. Indeed, Rh1 and Rg2 increased the degradation of Tim23, Tom40, and HSP60. These data suggested that ginsenoside Rh1 and Rg2 enhanced the mitophagy. After then, we clarified the effect of mitophagy on inflammasome activation. Mitophagy induced by Rh1 and Rg2 inhibited the inflammasome activation. Moreover, mtROS homeostasis plays an essential role in regulating autophagy, especially mitophagy [65]. Bacterial LPS is an important microbe-associated molecular pattern that induces various innate immune responses (such as ROS production) when sensed by the membrane-associated Toll-like receptor 4 [66,67]. Additionally, mtROS is involved in NLRP3 inflammasome activation. Mitochondrial complex I inhibitors enhanced mtROS production and induces IL-1β secretion, and this activation is blocked by ROS scavengers [24,68]. These results suggest that ginsenosides Rh1 and Rg2 promote mitophagy by alleviating mtROS production and maintain mitochondrial homeostasis to inhibit NLRP3 inflammasome activation.

## 5. Conclusions

In conclusion, saponin extract exerted antisteatotic, antifibrotic, and anti-inflammatory effects in the FFD-induced NAFLD mouse model. Saponin extract was found to be enriched in ginsenosides Rh1 and Rg2, which play a protective role against NAFLD by inhibiting the NLRP3 inflammasome, promoting mitophagy, and alleviating mtROS production. Thus, saponin extract administration may be used as an alternative therapeutic strategy for NAFLD.

## Figures and Tables

**Figure 1 nutrients-13-00856-f001:**
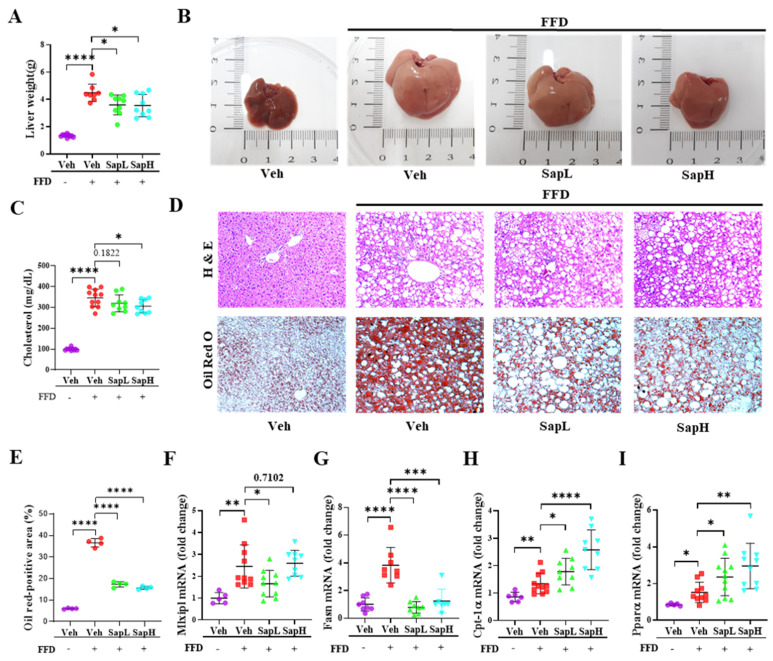
Saponin extract improved the fast food diet (FFD)-induced hepatic steatosis in vivo. Saponin extract was orally administered to mice for the last 9 weeks until the animals were sacrificed. (**A**) Liver weight gain was measured in the last week. (**B**) The appearance of the liver. (**C**) The level of cholesterol in the serum was measured. (**D**) Representative images of liver pathology, with H&E staining and Oil Red O staining; images were acquired at original magnification and 400×; scale bars, 20 μm. (**E**) The quantification of the Oil red positive area. Expression of (**F**) Mlxipl, (**G**) Fasn, (**H**) Cpt-1α, and (**I**) Pparα in liver tissue were measured by qRT-PCR and shown as folded changes compared with normal diet mice. Relative mRNA expression levels were normalized to mouse glyceraldehyde 3-phosphate dehydrogenase (GAPDH) levels. The data are expressed as means ± sem. * *p* < 0.05, ** *p* < 0.01, *** *p* < 0.001, and **** *p* < 0.0001.

**Figure 2 nutrients-13-00856-f002:**
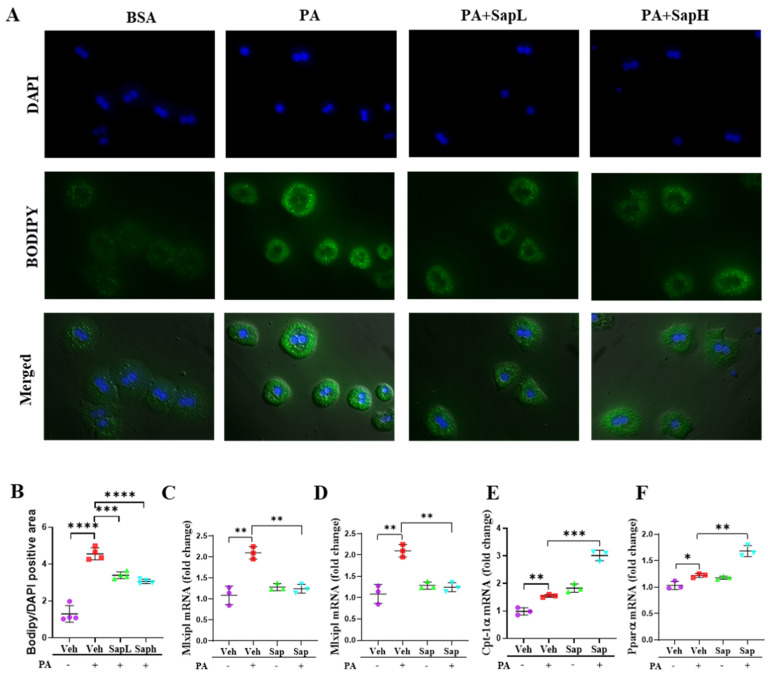
Saponin extract improved the FFD-induced hepatic steatosis in vitro. (**A**) Lipid accumulation was assessed by BODIPY staining in primary hepatocyte. (**B**) The quantification of the BODIPY/DAPI-positive area. Expression of lipogenesis genes (**C**) Mlxipl and (**D**) sterol regulatory element-binding protein (Srebp-1c) and fatty acid oxidation genes marker (**E**) Srebp-1c and (**F**) Pparα were measured by qRT-PCR in the primary hepatocyte. Relative mRNA expression levels were normalized to mouse GAPDH levels. The data are expressed as means ± sem. * *p* < 0.05, ** *p* < 0.01, *** *p* < 0.001, and **** *p* < 0.0001.

**Figure 3 nutrients-13-00856-f003:**
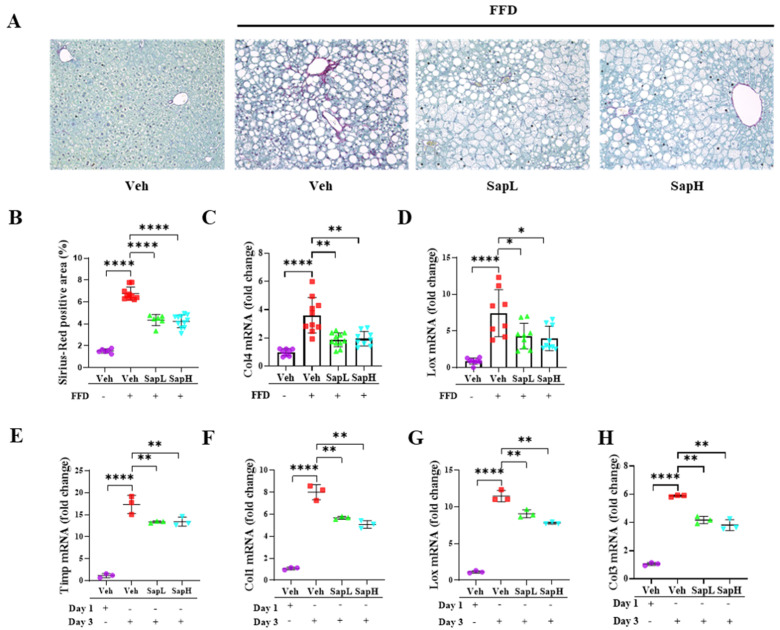
Saponin extract alleviated the FFD-induced steatofibrosis. (**A**) Representative photomicrographs of Sirius Red staining of mice liver. Collagen fibers were stained red. (**B**) Sirius red-positive area was calculated as a percentage of the total area. Expression of transcription factors associated with fibrosis, (**C**) collagen type IV (Col4) and (**D)** lysyl oxidase (Lox), in liver tissue were measured by qRT-PCR shown as a fold change compared with normal diet mice. The expression of fibrotic genes (**E**) tissue inhibitor of metalloproteinase (Timp), (**F**) Col1, (**G**) Lox, and (**H**) Col3 were measured by qRT-PCR in hepatic stellate cells (HSCs). Relative mRNA expression levels were normalized to mouse GAPDH levels. The data are expressed as means ± sem. * *p* < 0.05, ** *p* < 0.01, and **** *p* < 0.0001.

**Figure 4 nutrients-13-00856-f004:**
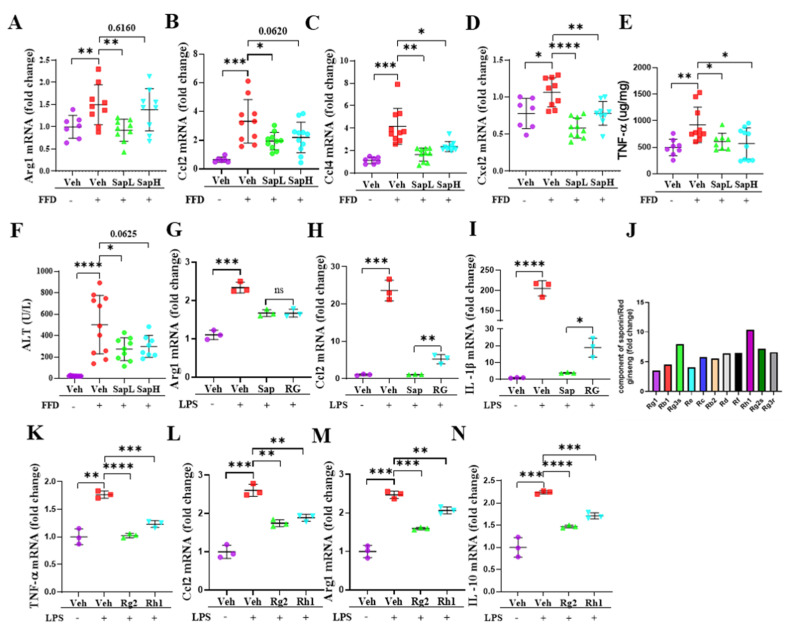
Saponin extract has high contents of ginsenoside Rh1 and Rg2 that exert an anti-inflammatory effect. The mRNA expression of inflammatory cytokines (**A**) Arg1, (**B**) Ccl2, (**C**) Ccl4, and (**D**) Cxcl2 were measured by qRT-PCR in the liver tissue. (**E**) The protein level of TNF-α in liver tissue lysates were determined by ELISA. (**F**) The level of ALT in the serum was measured. The mRNA expression of inflammatory cytokines in ImKCs, compared with saponin extract and red ginseng, (**G)** Arg1, (**H**) Ccl2, and (**I**) IL-1β. (**J**) The fold changes of different compounds in saponin extract and red ginseng. The mRNA expression of inflammatory cytokines in ImKCs, compared with ginsenoside Rh1 and Rg2, (**K**) TNF-α, (**L**) Ccl2, (**M**) Arg1, and (**N**) IL-10. Relative mRNA expression levels were normalized to mouse GAPDH levels. The data are expressed as means ± sem. * *p* < 0.05, ** *p* < 0.01, *** *p* < 0.001, and **** *p* < 0.0001.

**Figure 5 nutrients-13-00856-f005:**
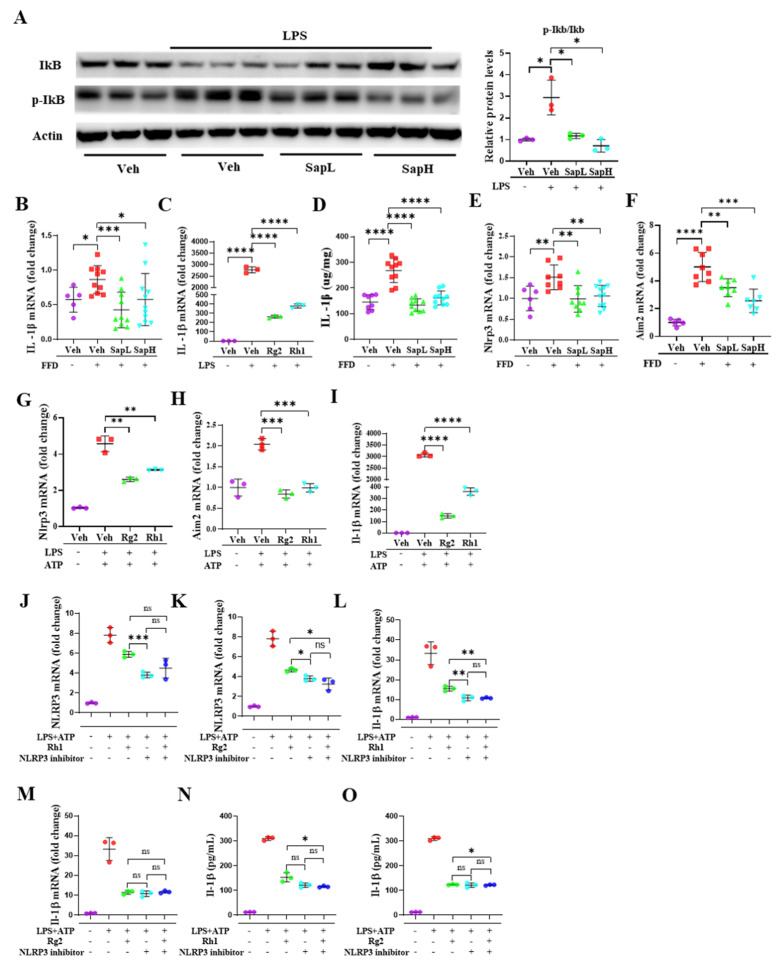
Ginsenoside Rh1 and Rg2 treatment inhibits the activation of the NLRP3 inflammasome. (**A**) Representative Western blotting analysis of IkB, p-IkB, and Actin. Saponin extract 62.5 µg/mL is referred to as SapH and Saponin extract 31.25 µg/mL as SapL. The mRNA expression of IL-1β in (**B**) liver tissue and (**C**) in ImKCs. (**D**) The protein level of IL-1β in liver tissue lysates was determined by ELISA. The mRNA expression of the NLRP3 inflammasome (**E**) and Aim2 inflammasome (**F**) in liver tissue and in ImKCs (**G**,**H**). The mRNA expression of IL-1β (**I**) in ImKCs. The mRNA expression of NLRP3 inflammasome treated with ginsenoside Rh1 (**J**) and Rg2 (**K)** in ImKCs. The mRNA expression of IL-1β treated with ginsenoside Rh1 (**L**) and Rg2 (**M**) in ImKCs. The protein level of IL-1β treated with ginsenoside Rh1 (**N**) and Rg2 (**O**) in the cell culture was determined by ELISA. Relative mRNA expression levels were normalized to mouse GAPDH levels. The data are expressed as means ± sem. * *p* < 0.05, ** *p* < 0.01, *** *p* < 0.001, and **** *p* < 0.0001.

**Figure 6 nutrients-13-00856-f006:**
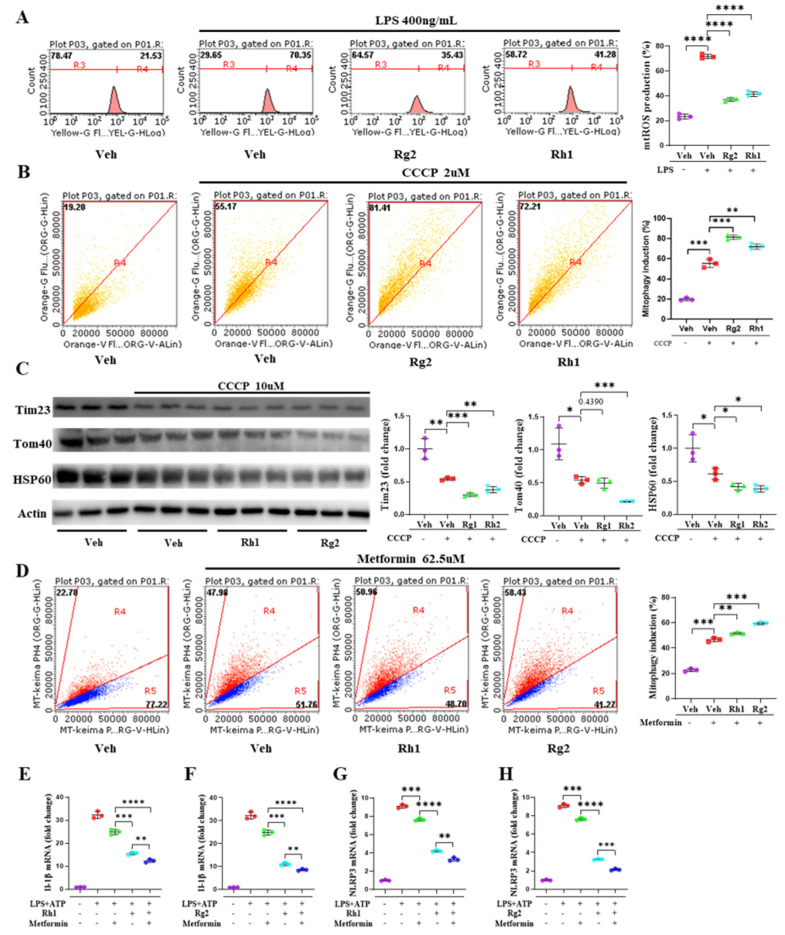
Ginsenoside Rh1 and Rg2 alleviated the activation of NLRP3 by promoting mitophagy. (**A**) ImKCs were measured for MitoSOX 3 h after treatment with LPS (500 ng/mL) and ginsenoside Rh1 (62.5 µg/mL) and Rg2 (62.5 µg/mL). (**B**) Hep3B cells were transfected with pLVX-puro-mt-Keima. Analysis of expressing mt-Keima treated with ginsenoside Rh1 and Rg2 for 24 h. Rh1 and Rg2 mediated mitophagy was quantified by flow cytometry with 2-µM carbonyl cyanide m-chlorophenylhydrazone (CCCP); the excitation wavelength was 586 nm in a neutral and 440 nm in an acidic environment. (**C**) Representative Western blotting analysis of Tim23, Tom40, and HSP60. The concentration of ginsenoside Rh1 and Rg2 was 62.5 µg/mL. (**D**) Ginsenoside Rh1 and Rg2-mediated mitophagy was quantified by flow cytometry with 62.5-µM metformin for 24 h in mt-Keima Hep3B cells. The mRNA expression of IL-1β in ImKCs cotreated with LPS+ATP, Metformin, and Rh1 (**E**) and Rg2 (**F**). The mRNA expression of the NLRP3 inflammasome in ImKCs cotreated with LPS+ATP, Metformin, and Rh1 (**G**) and Rg2 (**H**). Relative mRNA expression levels were normalized to mouse GAPDH levels. The data are expressed as means ± sem. * *p* < 0.05, ** *p* < 0.01, *** *p* < 0.001, and **** *p* < 0.0001.

**Table 1 nutrients-13-00856-t001:** Composition of the fast food diet.

Component	Western Diet (g)	Sugar Solution (g/L)
Methionine, DL	3	0
Lodex 10	100	0
Solka floc, FCC200	50	0
Corn oil	10	0
Calcium phosphate, dibasic	4	0
V10001C	1	0
Cholesterol, NF	1.5	0
Casein, lactic, 30 mesh	195	0
Sucrose, finely granulated	350	0
Starch, corn	50	0
Butter, anhydrous	200	0
S10001A	17.5	0
Choline bitartrate	2	0
Ethoxyquin	0.04	0
Glucose	0	18.9
Sucrose	0	23.1

DL: Methionine exists as D (dextrogyre) and L (levogyre) optical isomers, the racemic mixture of D and L-isomers forms DL-methionine, which is the commercially available methionine.

**Table 2 nutrients-13-00856-t002:** Primer sequences of the genes used for quantitative real-time polymerase chain reaction.

Gene	Forward (5′ to 3′)	Reverse (5′ to 3′)
GAPDH	ACGGCAAATTCAACGGCACAG	AGACTCCACGACATACTCAGCAC
MLXIPL	CTGGGGACCTAAACAGGAGC	GAAGCCACCCTATAGCTCCC
FASN	AGGTGCTAGAGGCCCTGCTA	AGGTGCTAGAGGCCCTGCTA
CPT-1α	CGGTTCAAGAATGGCATCATC	TCACACCCACCACCACGAT
PPARα	GCTACCACTACGCAGTTCACG	GCTCCGATCACAACTTGTCGT
SREBP-1c	GGAGCCATGGATTGCACATT	AGGAAGGCTTCCAGAGAGGA
COL4	AGGAGAGAAGGGTGAACAAGG	CCAGGAGTGCCAGGTAAGCC
TIMP1	TCTGGCATCTGGCATCCTCTTG	TCTGGCATCTGGCATCCTCTTG
LOX	GGTTACTTCCAGTACCGTCTCC	GCAGCGCATCTCAGGTTGT
COL1	TAGGCCATTGTGTATGCAGC	ACATGTTCAGCTTTGTGGAC
ARG1	CTCCAAGCCAAAGTCCTTAGAG	CTCCAAGCCAAAGTCCTTAGAG
CCL2	ATTGGGATCATCTTGCTGGT	CCTGCTGTTCACAGTTGCC
CCL4	CTCTGCGTGTCTGCCCTCTC	TGGTCTCATAGTAATCCATCAC
CXCL2	GCCAAGGGTTGACTTCAAGAACA	AGGCTCCTCCTTTCCAGGTCA
TNF- α	AGGGTCTGGGCCATAGAACT	CCACCACGCTCTTCTGTCTA
IL-1β	CTCGCAGCAGCACATCAACAAG	CCACGGGAAAGACACAGGTAGC
IL-10	GCTGGACAACATACTGCTAACCG	GCTGGACAACATACTGCTAACCG
NLRP3	ACTGAAGCACCTGCTCTGCAAC	AACCAATGCGAGATCCTGACAAC
AIM2	GGTTGATGTTGAATCTAACCACGAA	GGTTGATGTTGAATCTAACCACGAA

GAPDH, glyceraldehyde 3-phosphate dehydrogenase; MLXIPL, MLX-interacting protein-like; FASN, fatty acid synthase; CPT-1, carnitine palmitoyltransferase I; PPAR, peroxisome proliferator-activated receptor; SREBP, sterol regulatory element-binding protein; COL4, collagen type IV; TIMP, tissue inhibitor of metalloproteinase; LOX, lysyl oxidase; ARG, arginase; CCL, C-C motif ligand; CXCL, C-X-C motif ligand; TNF, tumor necrosis factor; IL, interleukin; NLR, nucleotide-binding domain and leucine-rich repeat-containing; and AIM, absent in melanoma.

**Table 3 nutrients-13-00856-t003:** Contents of the eleven ginsenosides in red ginseng (RG) and saponin extract.

Sample	Ginsenosides (mg/g)										
	Rg1	Rb1	Rg3s	Re	Rc	Rb2	Rd	Rf	Rh1	Rg2s	Rg3r
RG	0.64	4.15	2.43	0.69	1.61	1.41	0.65	0.92	0.81	1.1	0.91
Saponin extract	2.27	18.92	19.25	2.79	9.37	7.84	4.12	5.95	8.38	7.82	5.96

## Data Availability

The authors confirm that the data supporting the findings of this study are available within the article.

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
