# Peer review of "Ginseng Saponin Enriched in Rh1 and Rg2 Ameliorates Nonalcoholic Fatty Liver Disease by Inhibiting Inflammasome Activation"

_nutrients, 2021, doi:10.3390/nu13030856_

Round 1
Reviewer 1 Report
In this study, Wang et al. investigated the tole of ginseng saponin enriched in Rh1 and Rg2 in non-alcoholic fatty liver disease. By providing both in vivo and in vitro evidence, the authors concluded that ginsenosides ameliorates non-alcoholic fatty liver disease by inhibiting inflammasome activation and promoting mitophagy. The authors provides convincing evidence suggesting a role of ginsenosides in preventing disease pathogenesis, however, there are still some missing link between the proposed molecular mechanisms.
- In Fig5, the rt-pcr results show that NLRP and Aim2 mRNA level is decreased upon ginsenosides treatment. The authors concluded that Rh1 and Rg2 suppress the LPS-ATP induced activation of NLRP3 and AIM2 inflammasomes (line 273). Since AIM2 inflammasome also involves in IL-1b processing, it is reasonable to ask whether AIM2 inflammasomes are activated in the disease pathogenesis and whether ginsenosides treatment inhibit both inflammasome activation pathways.
- In figure 6D-H, the authors should use an inhibitor of mitophagy, but not an activator of mitophagy to determine whether ginsenosides alleviate NLRP3 activation by promoting mitophagy.
- It seems that the link between inhibition of inflammasome signaling and the enhancement of mitophagy is missing. Do these two events occur simultaneously? Or is mitophagy activation a downstream effect of inflammasome signaling? The authors need to address this issue.
Reviewer 2 Report
The manuscript showed interesting study results of ginseng saponin and its ginsenosides effects on hepatic steatosis, fibrosis, and inflammation and their underlying action mechanism in non-alcoholic fatty liver disease (NAFLD). However, few significant issues need to be addressed:
Page 1; Line 9: Abstract needs to be constructed in a way that all the outcomes with the significant figures should be given either in the percentage or statistically significant values.
Page 3; Line 89: It would be great if a separate section as "Chemicals" should be included which can show all the chemicals, assay kits used in the study.
Page 3; Line 107: What was the initial passage of the cells?
Page 5; Line 176: What the number of the cells/well as given in per mL volume.
One of the major issues in the manuscript was the absence of figures which were only written in text.

Reviewer 3 Report
Interesting experiment even if ginseng activities on NAFLD is not really new. The main problem with this draft is that we absolutly do not know what extract or pure molecules have been tested. Confusion is maintained between what is called saponin, red ginseng and Rh1 and Rg2. In the methods descriptions, it is a serious lack, the non-description of the obtention of the red ginseng, the saponin and the Rh1 and Rg2. Are saponin an extract ? And how was it obtained ? Who's the supplier of the pure Rh1 and Rg2, the described experiment present these molecule tested as pure. Please used latine name for the plant. The main difficulty of this draft is that we never understand if you made a focus on two specific ginsenosides present in a extract (Table 3) because the other ginsenosides present in the extract have been already tested for NAFLD, or if pure ginsenosides have been tested in comparison with the extract described in table 3. You must clarify this point please.
Round 2
Reviewer 1 Report
None.
Author Response
Please check the manuscript. Thank you!
Reviewer 3 Report
Dear authors, thanks for answers and explanation, and draft improvement.
In the added redaction in red, may I suggest to not use CCCP-treated group, but CCCP-treated cells. Group is useful for animal experiment and CCCP was used to treat cells. Same comment when cells treated with Rh1 and Rg2.
Please improve English in the added redaction.
In the introduction and in the text, the redaction could be improved by clarifying what is called saponin, that is the extract described now in 2.3. Saponin is a general meaning word that could be confused with your extract obtained according to your method. In 2.3 you called it saponin extract, so it could be more rigorous to replace saponins by saponin extract when dealing with your extract.
